**www.cambridge.org/qrd**

# Amyloid-like Hfq interaction with single-stranded DNA: involvement in recombination and replication in *Escherichia coli*

**Key words:**
bacterial amyloid; Hfq; non-coding RNA; nucleoid associated protein; single-stranded DNA-binding protein; Sm-like protein

**Author for correspondence:**
*Véronique Arluison,
E-mail: veronique.arluison@u-paris.fr;
Grzegorz Wegrzyn,
E-mail: grzegorz.wegrzyn@biol.ug.edu.pl

Krzysztof Kubiak[1,2] , Frank Wien[3] , Indresh Yadav[4], Nykola C. Jones[5] ,
Søren Vrønning Hoffmann[5] , Eric Le Cam[6], Antoine Cossa[2,7] ,
Frederic Geinguenaud[8] , Johan R. C. van der Maarel[4] , Grzegorz Węgrzyn[1]* and
Véronique Arluison[2,9]* 

[1]Department of Molecular Biology, University of Gdansk, Wita Stwosza 59, 80-308 Gdansk, Poland; [2]Laboratoire Léon Brillouin, Université Paris Saclay, CEA, LLB, 91191 Gif-sur-Yvette, France; [3]DISCO Beamline, Synchrotron SOLEIL, 91192 Gif-sur-Yvette, France; [4]Department of Physics, National University of Singapore, Singapore 117542, Singapore; [5]ISA, Department of Physics and Astronomy, Aarhus University, 8000 Aarhus C, Denmark; [6]UMR9019-CNRS, Genome Integrity and Cancer, Université Paris-Saclay, Gustave Roussy, F-94805 Villejuif Cedex, France; [7]Institut Curie, PSL University, Université Paris-Saclay, UMS2016, Inserm US43, Multimodal Imaging Centre, 91400 Orsay, France; [8]Plateforme CNanoMat and Inserm, U1148, Laboratory for Vascular Translational Science, UFR SMBH, Université Paris 13, Sorbonne Paris Cité, F-93017, Bobigny, France and [9]Université Paris Cité, UFR SDV, 75006 Paris, France

## Abstract

Interactions between proteins and single-stranded DNA (ssDNA) are crucial for many fundamental biological processes, including DNA replication and genetic recombination. Thus, understanding detailed mechanisms of these interactions is necessary to uncover regulatory rules occurring in all living cells. The RNA-binding Hfq is a pleiotropic bacterial regulator that mediates many aspects of nucleic acid metabolism. The protein notably mediates mRNA stability and translation efficiency by using stress-related small regulatory RNA as cofactors. In addition, Hfq helps to compact double-stranded DNA. In this paper, we focused on the action of Hfq on ssDNA. A combination of experimental methodologies, including spectroscopy and molecular imaging, has been used to probe the interactions of Hfq and its amyloid C-terminal region with ssDNA. Our analysis revealed that Hfq binds to ssDNA. Moreover, we demonstrate for the first time that Hfq drastically changes the structure and helical parameters of ssDNA, mainly due to its C-terminal amyloid-like domain. The formation of the nucleoprotein complexes between Hfq and ssDNA unveils important implications for DNA replication and recombination.

## Introduction

Although most genetic information is stored in double-stranded (dsDNA), genetic expression requires unwinding of DNA into single-stranded DNA (ssDNA). In particular, transient portions of ssDNA appear during replication, recombination or repair processes; ssDNA is more sensitive to nuclease degradation and this leads to the formation of secondary structures, which prevent previously cited processes. To solve these problems, cells need specialised ssDNA-binding proteins (SSB) that bind and stabilise ssDNA structures. These proteins usually do not share significant sequence similarity, but all contain a DNA-binding oligo-nucleotide binding OB fold (Su *et al.,* 2014), consisting of a five-stranded curved β sheet capped by a helix. This specific fold is responsible not only for ssDNA binding but also often for the self-assembly of oligomeric SSB. In bacterial cells, ssDNA fragments also play crucial roles, including nucleoid and phage DNA replication or genetic recombination (Molan and Zgur Bertok, 2022).

In bacterial chromosome replication, ssDNA regions are necessary for initiating the process of DNA synthesis by melting the double helix, recruiting the multiprotein machinery and allowing formation of primers and replication forks (Zawilak-Pawlik *et al.,* 2017). Therefore, proteins interacting with ssDNA and/or stabilising such DNA fragments, like the single-stranded DNA-binding protein, play important roles in facilitating the formation of nucleoprotein complexes at replication forks and maintaining their functions (Oakley, 2019). Bacterial proteins that interact with such DNA structures particularly affect replication of viruses, including bacteriophages, having genomes composed of ssDNA (Shulman and Davidson, 2017). The transition from ss to dsDNA is a crucial step in propagation of such viruses; thus, proteins interacting with ssDNA can significantly modulate the viral replication processes.

Another process in which ssDNA is crucial is homologous recombination. DNA strand displacement and replacement occurring in this process require the formation of complexes

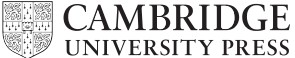

including ssDNA and specific proteins (Piazza and Heyer, 2019), and the influence of ssDNA-interacting proteins may be important in modulating recombination efficiency. The function of the bacteriophage λRed-recombination system may serve as an example of such an ssDNA-dependent process (Sharan *et al.,* 2009). In summary, proteins interacting with ssDNA may influence various biological processes, and thus, one should consider that any ssDNA-binding protein might modulate chromosomal and viral replication, as well as genetic recombination.

In this work, we focused our attention on the ssDNA-binding property of the *Escherichia coli* Hfq protein. Hfq is an abundant protein that flexibly binds nucleic acids (NA) (Rajkowitsch and Schroeder, 2007; Vogel and Luisi, 2011). Structurally, the amino-terminal region of Hfq (NTR, 65 residues) shares homologies with the Sm family of protein (Wilusz and Wilusz, 2013) adopting the OB-like fold. Hfq-NTR is comprised of five β-strands and another Sm-proteins assembles into a cyclic oligomer to form the functional unit (Brennan and Link, 2007). Although the mechanism by which Hfq binds NA is not completely clear, it is now established that its NTR binds RNA and DNA. Uridine-rich RNA is bound to one face of the torus called the proximal face, while the A-rich sequences bind to the opposite distal face; dsDNA is also bound on the proximal face of the ring (Supplementary Fig. S1) (Link *et al.,* 2009; Orans *et al.,* 2020).

In addition to the well-characterised Sm domain, the Hfq protein also possesses a C-terminal region (CTR, 40 residues) located at the periphery of the torus (Arluison *et al.,* 2004). Although no atomic 3D structure is known for the CTR, it has been shown to self-assemble into an amyloid structure (Fortas *et al.,* 2015). Recently, it has been suggested that the CTR collaborates with the different RNA binding faces of Hfq, with important outcomes for some RNAs (Kavita *et al.,* 2022).

Functionally, Hfq controls a large number of bacterial functions. Among them, most are related to RNAs. Hfq was first identified as a host factor for a RNA bacteriophage but later found to play a general role in RNA metabolism (Vogel and Luisi, 2011). In particular, it facilitates the pairing of small non-coding RNA (sRNA) with target mRNA, allowing gene regulation at the post-transcriptional level. Indeed, Hfq allows a tight and fast regulation of gene expression and triggers stress-relief pathways (Gottesman, 2019).

Interestingly, Hfq also binds to DNA (Cech *et al.,* 2016) and some of the phenotypic effects appearing due to the lack of Hfq may be linked to defects in DNA-related processes. Hfq binds both linear and circular dsDNA (Takada *et al.,* 1997) and a significant amount of the protein is found in the nucleoid (~20%) (Diestra *et al.,* 2009). Hfq binding results in the condensation of DNA through protein–protein interactions. This activity, consisting of the mediation of NA interactions and referred to as bridging, is observed for other nucleoid-associated proteins (NAPs) to form loops (Rajkowitsch and Schroeder, 2007; Wiggins *et al.,* 2009).

The compaction of DNA by Hfq *in vitro* is mainly due to its CTR (Malabirade *et al.,* 2017a). Thus, Hfq probably plays a critical role in the architecture of the chromosome, even if this has not been established formally. If Hfq does not affect DNA supercoiling and transcription directly (Malabirade *et al.,* 2018), it possibly regulates them indirectly, for instance by regulating the expression of a transcriptional regulator (Majdalani *et al.,* 1998).

The work reported here further explores a newly discovered property of Hfq, its ability to bind ssDNA and to drastically change ssDNA structure by promoting its alignment. Effects of Hfq on DNA transactions in which ssDNA regions are crucial have been assessed.

## Methods

Details of methods can be found online as Supplementary Material.

### Protein expression and purification

Wild-type (WT) and truncated Hfq forms of *E. coli* Hfq were purified as described previously (Taghbalout *et al.,* 2014; Malabirade *et al.,* 2017b). CTR peptide was chemically synthesised. This part of the protein cannot be purified from bacteria as it is unstable when translated alone (Taghbalout *et al.,* 2014). We determined that the pH ~ 5 used was the most appropriate to form the complex with DNA. Although pH 5 seems to be far from physiological conditions, it is still relevant as Gram-negative bacteria can acidify their cytosol when adapting to the vacuoles of the host macrophage, and many virulent genes belong to the Hfq regulon in these bacterial species (Kenney, 2019).

### Fluorescence anisotropy measurements

Fluorescence anisotropy measurements were collected as described previously (Geinguenaud *et al.,* 2011).

### Optical microscopy of ssDNA-Hfq/CTR/NTR complexes

DNA in the single-stranded form was prepared by alkali-induced denaturation of double-stranded λ-DNA (Basak *et al.,* 2019). Then, Hfq (29.8 μM) was added to the solution with concentration of one hexamer-Hfq per 200 bases. A similar procedure was used to make the ssDNA with Hfq-CTR and Hfq-NTR. However, the molar concentration of CTR used was six times higher than Hfq, that is, 6 CTR for 200 bases of DNA. For fluorescence imaging, 1 h before imaging, ssDNA was stained with YOYO-1 at a concentration of one YOYO-1 dye per four bases.

### Synchrotron radiation circular and linear dichroism

Complex between $dA_{59}$ and Hfq-CTR was prepared as described previously (El Hamoui *et al.,* 2020). Synchrotron radiation circular dichroism (SRCD) measurements were carried out at DISCO/SOLEIL Synchrotron (proposal 20200007). Samples were loaded into a $CaF_2$ circular cell (24 μm path length). Due to the origin of absorption, spectra of mixed samples could not be standardised to $\Delta\varepsilon$ and spectra are presented in mdeg maintaining the same molar ratios for all presented samples. Synchrotron radiation linear dichroism (SRLD) measurements were carried out in the same cell by collecting triplicates every 90° from 0 to 270°. For the data acquisition, the modulator phase was set to $\lambda \times$ 0.608 doubling the lock-in amplifier frequency in order to measure only LD absorption.

### Couette flow SRLD

Couette flow SRLD measurements were performed at the AU-CD beamline on the ASTRID2 Synchrotron (proposal ISA-21-102), as detailed in Wien *et al.* (2019). Due to the much larger path length of the Couette flow cell used for SRLD measurements (0.5 mm) compared to the path length used for SRCD measurements

(0.024 mm), the complex between CTR and the $dA_{59}$ had a very strong LD signal under flow conditions; the samples were diluted (1/36) compared to the concentrations used for SRCD.

### Fourier transform infrared spectroscopy

For Fourier transform infrared (FTIR) analysis, the same solutions used for SRCD analysis were lyophilised and re-dissolved in $D_2O$ (5 µL). FTIR measurements were performed as detailed in Geinguenaud et al. (2011).

### Transmission electron microscopy imaging

3.65 nM circular ssDNA molecules extracted from ΦX174 virions was incubated with 100 nM CTR in Tris–HCl 10 mM pH 7.5 EDTA 1 mM for 10 min at 20 °C. 5 µl was then deposited onto positively functionalised grids covered with a thin carbon film (Beloin et al., 2003). Grids were washed with aqueous 2% (w/v) uranyl acetate, dried and observed in annular darkfield mode using a Zeiss 902 EM. Veletta CCD camera is controlled by iTEM software (Olympus Soft Imaging).

### E. coli strains and bacteriophages

E. coli WT strain MG1655 (Jensen, 1993) was used as the $hfq^+$ positive control. Its Δhfq and ΔCTR derivatives were constructed as described in Gaffke et al. (2021). Quantification of Hfq and its CTR-truncated form was by Western blotting and confirmed by dot blotting. For propagation of bacteriophage M13, $hfq^+$, Δhfq and ΔCTR derivatives of E. coli strain Hfr3000 (Bachmann, 1972) were constructed by P1 transduction. Bacteriophages M13 (Salivar et al., 1964), λcI857S7(am) (Goldberg and Howe, 1969), called λS(am) in this work, and λb519imm21susP (Wegrzyn et al., 1995), called λP (am) in this work, were used. E. coli strain TAP90 (Patterson and Dean, 1987) was used for propagation and titration of phages λcI857S7(am) and λb519imm21susP.

### Bacteriophage M13 development and efficiency of phage λ recombination

Development of phage M13 phage and λ recombination efficiency were tested according to the previously described method (Mosberg et al., 2010).

## Results

### Hfq binds, coats and spreads ssDNA

The potential of Hfq to bind to ssDNA was investigated. We chose $(dA)_n$ sequences because Hfq has the highest affinity for A-rich sequences (Folichon et al., 2003; Geinguenaud et al., 2011). Indeed, A-tracts are over-represented and distributed throughout the E. coli genome in phased A-repetitions (~ 12 nucleotides), organised in approximately 100 nucleotides-long clusters (Tolstorukov et al., 2005). Titrations of $dA_7$, $dA_{20}$ and $dA_{59}$ with Hfq gave $K_d$ values of $3.5 \pm 0.2$ µM, $183 \pm 8$ nM and $166 \pm 16$ nM, respectively (Supplementary Fig. S2). A weaker affinity was measured for the CTR and $dA_{59}$, about $4.2 \pm 0.3$ µM, thus suggesting that Hfq can bind $dA_{59}$ using both its NTR and CTR regions, as previously observed for longer dsDNA (Jiang et al., 2015).

The coating of ssDNA by Hfq and its truncated forms was then tested. For this purpose, different approaches have been used. First DNA in its single-stranded form was prepared by alkali-induced denaturation of λ dsDNA. The ssDNA molecules were subsequently complexed with the protein following a buffer exchange according to the method previously reported (Basak et al., 2019). Prior to imaging using fluorescence microscopy, the complexes were stained with YOYO-1 dye. The fluorescence intensity is relatively weak, because for ssDNA the dye is side-bound and cannot intercalate. Two different types of experiments have been done: stretched on a surface combed and confined in a nanochannel. First, the coated molecules were analysed on a flat surface (Allemand et al., 1997). The images are shown in Fig. 1a; the corresponding measured average stretches are also presented. As shown, the results are almost identical for Hfq, CTR or NTR. The ssDNA molecules are combed to a similar length of $25 \pm 2$ µm (0.52 nm/base), irrespective of the protein or its truncated forms. This confirms that both the NTR and CTR bind and coat ssDNA. A second type of experiment was done using a nanofluidic device to mimic a confined environment. In this experiment, ssDNA coated with Hfq was brought inside a rectangular channel with a cross-sectional diameter of 125 nm using an electric field. Once the complexes are inside the channel, they were allowed to relax for 2 min before imaging. The stretch of the complexes in the longitudinal direction of the channel was measured and is shown in the histogram in Fig. 1b. The corresponding histogram pertaining to the combing experiment has also been included. The mean stretch of the nano-confined complexes was found to be $15 \pm 2$ µm. In a rigid origami structure, ssDNA takes a maximal stretch $0.68 \pm 0.02$ nm/base (Roth et al, 2018). Accordingly, the stretch per unit contour length (relative stretch) of the complexes inside the channel amounts $0.46 \pm 0.06$. In the case of naked dsDNA confined to the same channel, the relative stretch was reported to be $0.528 \pm 0.005$ (Yadav et al., 2020). It must be emphasised that bare ssDNA without protein coating is notoriously difficult to linearise with molecular combing and cannot be stretched inside a nano-channel due to strong intramolecular hybridisation. We conclude that the relative stretches of ssDNA coated with full-length Hfq and naked dsDNA confined to the 125 nm channel are similar, which indicates that the bending rigidity (persistence length) of the coated ssDNA filament is about the same as that of bare dsDNA. In contrast, previous reports with another method allowing imaging of naked ssDNA pointed to lower values with a wide variation for ssDNA extension, that is, from 0.18 nm/base to 0.36 nm/base (Hansma et al., 1992; Woolley and Kelly, 2001), thus ssDNA extension by Hfq and its CTR (0.52 nm/base) is significantly more. Extension of ssDNA by Hfq and/or its CTR also exceeds the extension following coating of ssDNA with a cationic-neutral polypeptide copolymer (0.32 nm/base; Basak et al., 2019). Note that in these experiments, the coated ssDNA molecules are aligned through the application of flow (combing) or confinement to a long and narrow channel. Although some intermolecular aggregation was observed, these aggregates generally break up due to the alignment procedure. Quantitative information regarding intermolecular bridging could, hence, not be obtained from these essentially single-molecule stretching experiments.

The effect of Hfq-CTR was then confirmed with Transmission electron microscopy (TEM). This experiment was done with circular viral DNA (single-stranded) and not linear molecules (Fig. 1c). Circular ssDNA molecules were deposited on a carbon surface previously functionalised with positive charges and positively stained. TEM clearly confirms the capability of CTR to spread some regions of the viral DNA in conjunction with folding of other regions by intra- or intermolecular bridging interaction (Fig. 1c, sub-panels b1 and b2). Note that the relative alignment, parallel or

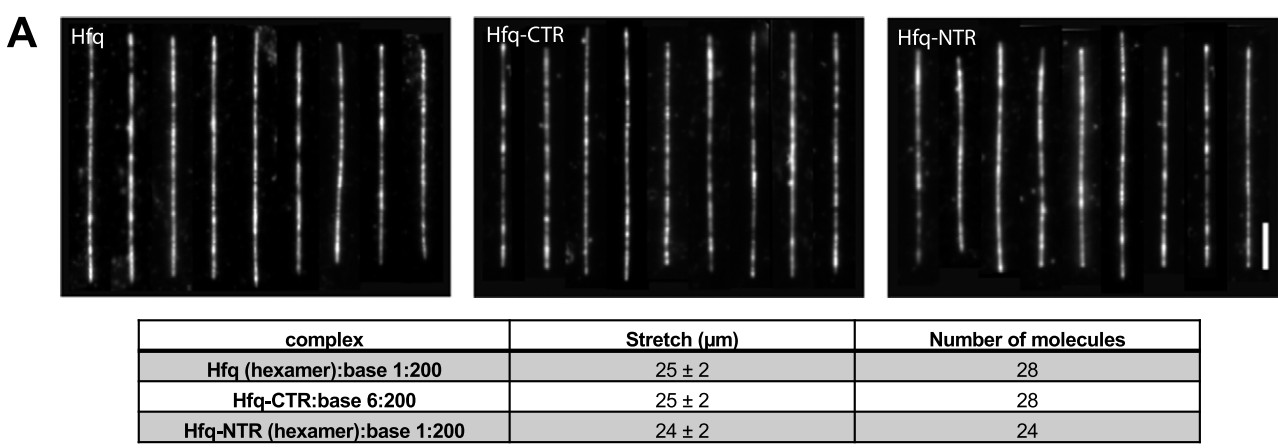

| complex | Stretch (μm) | Number of molecules |
|---|---|---|
| Hfq (hexamer):base 1:200 | 25 ± 2 | 28 |
| Hfq-CTR:base 6:200 | 25 ± 2 | 28 |
| Hfq-NTR (hexamer):base 1:200 | 24 ± 2 | 24 |

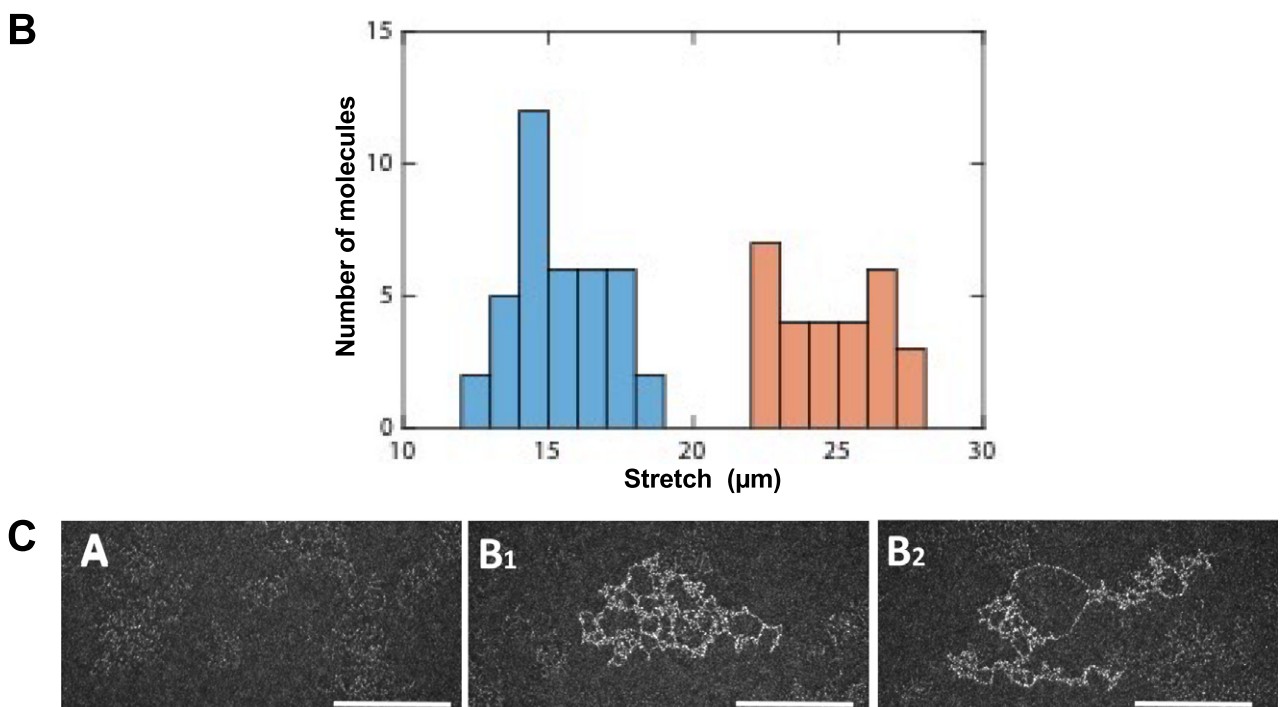

**Fig. 1.** ssDNA coating by Hfq. (*a*) Montages of ssDNA coated with Hfq or its truncated forms. Molecules are stained with YOYO-1 and stretched on the flat surface by molecular combing. *Scale bar* 5 μm. The corresponding measured extension for combed molecules is also given. Note that the molar concentration of CTR is kept six times higher than hexameric-HFq to maintain the stoichiometric ratio. (*b*) Histogram of ssDNA molecules imaged in combing (orange, 28 molecules) and inside nanofluidic channel (blue, 39 molecules). The average extension for the combing experiment is 25 ± 2 μm and in the nanofluidic channel is 15 ± 2 μm. (*c*) TEM evidence of Hfq-ssDNA binding. ΦX174 ssDNA virions were incubated in the presence of CTR. Before incubation, virions ssDNA is difficult to visualise (sub-panel *a1*). CTR binding to ssDNA ΦX174 allows the spreading of some region of the DNA, while others are strongly bridged (sub-panels *b1* and *b2*). In this case, the CTR causes the association of several ΦX174 that cannot be differentiated and the length of the viral DNA cannot be measured. *Scale bars:* 200 nm.

antiparallel, of ssDNA portions in the bridged regions cannot be determined.

### *Hfq changes ssDNA structure and allows DNA alignment*

Next, the effect of Hfq-CTR on ssDNA has been investigated using SRCD. We identified significant spectral changes over the whole spectrum (Fig. 2). Assuming that the protein is restructuring (amyloid formation) upon DNA binding (Malabirade *et al.,* 2018), we also identified that the ssDNA structure changes. We note a positive band at 180 nm and a negative band at 190 nm both indicative of left-handed NA (Wien *et al.,* 2021). In contrast, a positive band around 185 nm and a negative one between 200 and 210 nm are indicative of right-handed NAs (Wien *et al.,* 2021). The

spectral change identified could be correlated with base-tilting of A-rich sequences (Edmondson and Johnson, 1985; Wien *et al.,* 2019). Furthermore, the influence of Hfq on the positive CD signal around 270 nm may be influenced by base pairing and stacking (Holm *et al.,* 2010; Wien *et al.,* 2021).

One possible explanation for the spectral inversion observed could be due to a change in helical parameters. To test this possibility, we used FTIR spectroscopy to analyse sugar puckering. When only C3'-endo sugars are present, characteristic absorption bands located at 865 and 820 cm$^{-1}$ are observed, corresponding to the A-form. A contribution around 840 cm$^{-1}$ is indicative of a C2'-endo conformation (B-form) (Wien *et al.,* 2021). The formation of a Z-like form is excluded as it would give a triplet at 970, 951 and 925 cm$^{-1}$ (Banyay *et al.,* 2003). The A-form would give a triplet

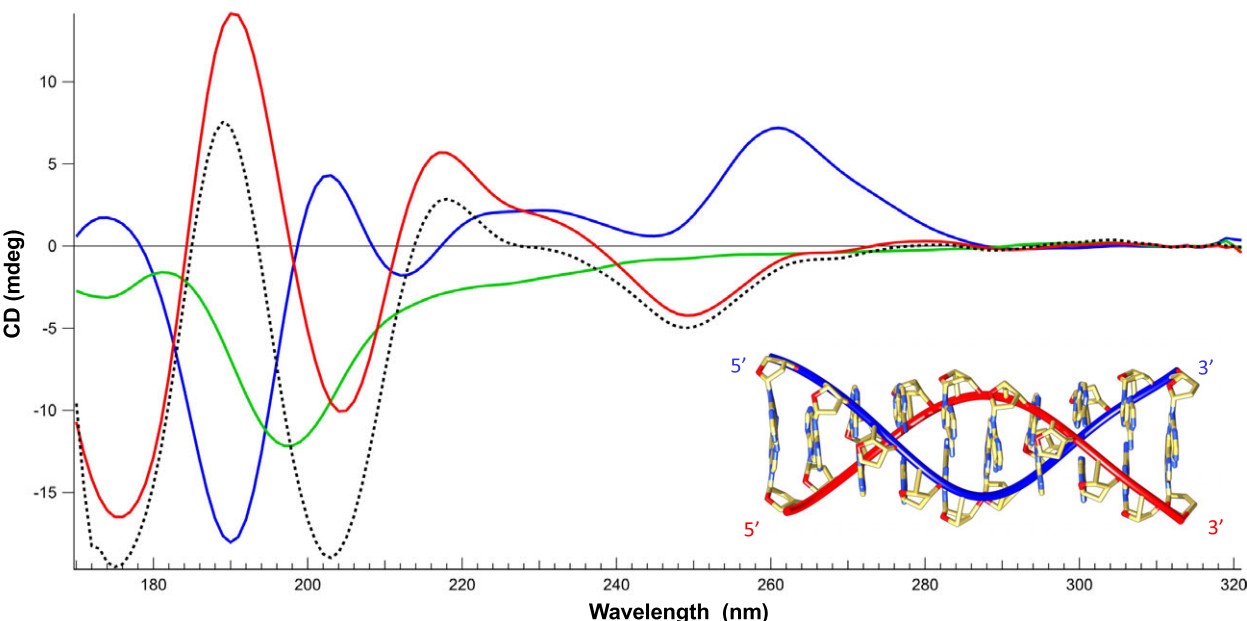

**Fig. 2.** Structure characterisation of ssDNA complexed to Hfq-CTR by SRCD spectroscopy. Spectra of DNA in the absence (red) and presence of CTR (blue). CTR alone (green). The dotted spectrum represents the theoretical sum of individual spectra of the DNA and CTR. The measured spectrum of the complex is significantly different compared to the DNA + peptide theoretical spectra, indicating an conformational change of the complexed ssDNA. Note that the same analysis with the full-length protein was impractical due to the low solubility of the protein. Inset: model of parallel DNA (Gleghorn *et al.,* 2016).

at 977, 968 and 952 cm$^{-1}$ and the B-form would give a singlet at 970 cm$^{-1}$. As shown in Fig. 3*a*, we clearly see that the d-ribose stays in C2'-endo since we observe the typical bands near 840 and 970 cm$^{-1}$. We thus conclude that ssDNA complexed to Hfq remains in B-form.

A possibility could be that the pH 5 used in our conditions could result in the formation of A$^+$ adenine and subsequently in a parallel helix (Gleghorn *et al.,* 2016). FTIR analysis of our complex confirms this hypothesis: the band at 1658 cm$^{-1}$ is absent in dA$_{59}$ alone, but the shift from 932 to 947 cm$^{-1}$ and the net decrease of the band intensity at 1080 cm$^{-1}$ show that a parallel helix is formed by dA$_{59}$ when bound to CTR (Fig. 3*b*) (Taillandier and Liquier, 2002). This result is confirmed using SRCD as dA$_{59}$ complexed to CTR exhibits a spectrum similar to that of poly(dA) parallel helix (Holm *et al.,* 2012). We thus conclude the CTR induces the formation of ds parallel helix from ss dA$_{59}$. This is not the case for ssDNA alone; thus, the CTR promotes the formation of this parallel helix.

Next, as the formation of such a parallel helix could result in the alignment of ssDNA, we analysed the possible alignment of the ssDNA molecule by CTR. Indeed, we previously observed that the CTR amyloid region has a propensity to align DNA (Wien *et al.,* 2019). As shown in Fig. 4, a clear alignment by the CTR:ssDNA complex can also be observed under Couette flow conditions. The alignment without rotation in a flow cell was also confirmed by SRLD using a CD cell and a CD cell rotation chamber (Fig. 4*c*). Alignment of the complex in the CD sample may occur upon loading between short path lengths; hence, linear dichroism (LD) signals emerge. These signals, if strong enough, spill into the CD signal via LD-CD cross talk (Sutherland, 2009) and distort the CD spectrum. In order to eliminate the LD contribution acquisition of CD spectra at four angles (0°-90°-180°-270°) were averaged.

Shown in Fig. 4*a* are the SRLD spectra of CTR, ssDNA and finally of the complex CTR:dA$_{59}$. The LD signal is effectively zero for both Hfq-CTR and ssDNA compared to the very strong LD signal of the complex. This shows that although the two components do not align under flow conditions, the complex readily aligns. In contrast to the LD signal observed for dsDNA complexes with Hfq-CTR (Wien *et al.,* 2019), we observe that the LD signal is positive for all wavelengths including the wavelength range from 240 to 300 nm where the DNA signal dominates. This leads to a surprising conclusion: where long DNA normally aligns along the flow direction giving rise to negative LD signals (Dicko *et al.,* 2008), the parallel dsDNA formed by dA$_{59}$ are incorporated into the polymers of the complex in such a way that they are aligned perpendicular to the length of the amyloid strand. The positive signal at 195 nm is in agreement with the β-sheet secondary structure aligned perpendicular to the amyloid fibrils and thus the flow (Wien *et al.,* 2019) since β-strands have a transition dipole moment near 195 nm aligned perpendicular to the direction of the strand (Nordén *et al.,* 2010).

In conclusion, our results clearly show that *in vitro* the Hfq protein and in particular its CTR amyloid region are able to bind to ssDNA, cover, spread and bridge it, and to allow the alignment of ssDNA molecules. Because ssDNA is an essential intermediate in many DNA metabolic processes, we then analysed the effect of Hfq *in vivo* for two processes where ssDNA is formed. Hfq could indeed be part of the ssDNA-binding SSB protein family in prokaryotic cells, which bind to ssDNA, stabilising and protecting intermediate states of DNA recombination and replication.

This could particularly be the case for A-rich sequences found throughout the *E. coli* genome and when pH decreases during host infection (Tolstorukov *et al.,* 2005; Kenney, 2019). Furthermore, it could prevent ssDNA chemical attack or be involved in removal of

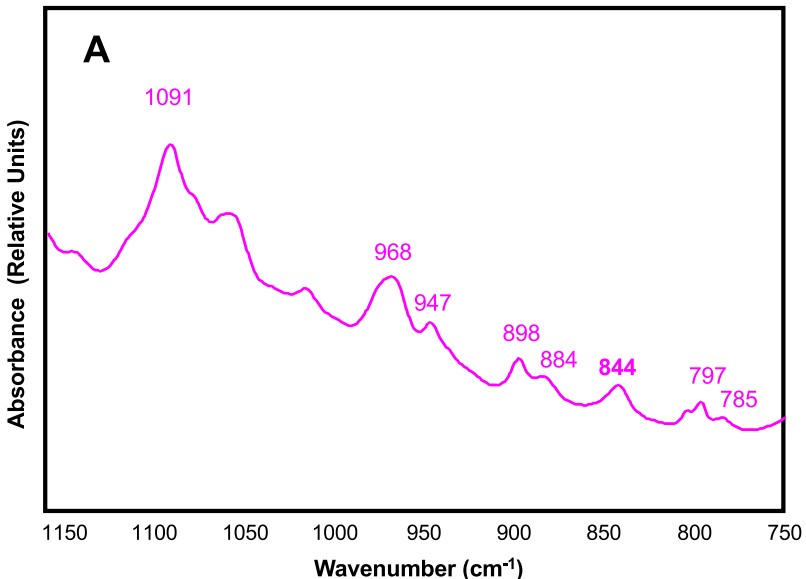

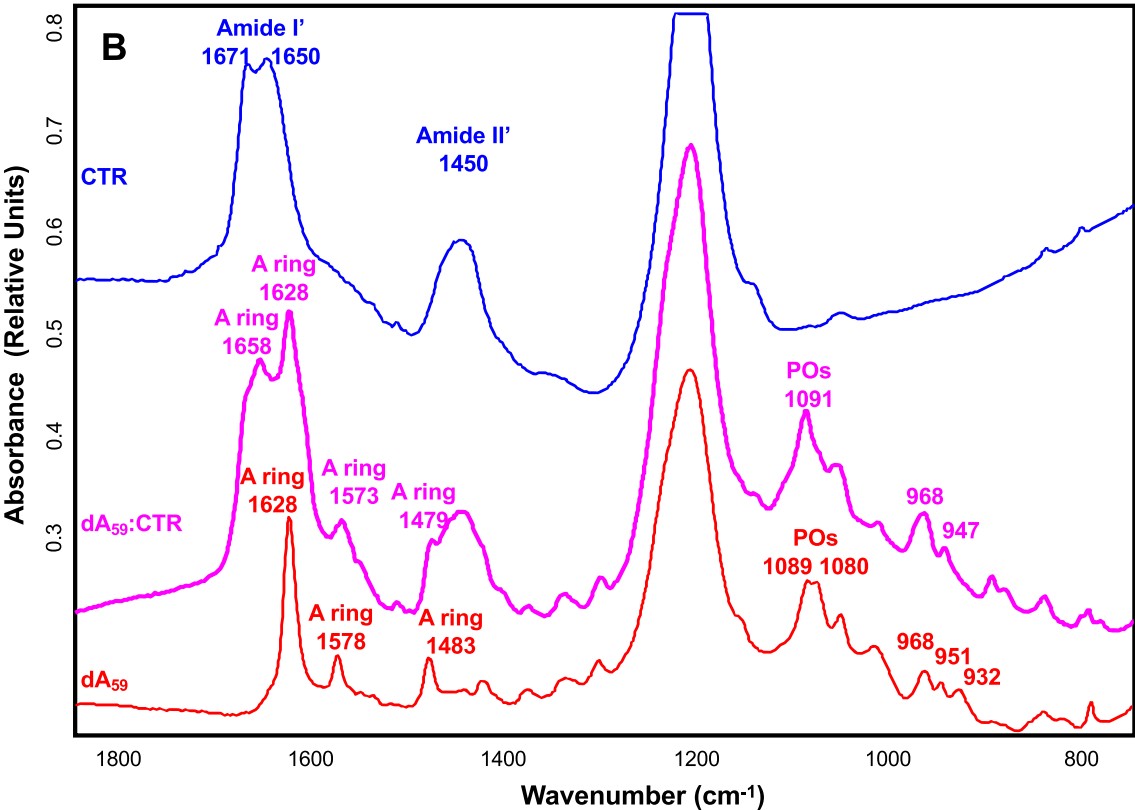

**Fig. 3.** (*a*) FTIR transmission spectrum of ssDNA in the presence of Hfq-CTR. The ribose stays in C2'-endo since we observe the typical bands at 840 and 970 cm$^{-1}$. (*b*) FTIR transmission spectrum of ssDNA in the presence of CTR. The band observed at 1658 cm$^{-1}$, absent in dA$_{59}$ alone, the shift from 932 to 947 cm$^{-1}$ and the net decrease of the band intensity at 1089 cm$^{-1}$ indicates that a parallel helix is formed by dA$_{59}$ when bound to CTR.

secondary structures that could impair ssDNA-related processes (Molan and Zgur Bertok, 2022).

### Hfq influences replication of bacteriophage M13 and genetic recombination

To evaluate if Hfq influences biological processes in which ssDNA regions are involved, we have tested efficiencies of M13-bacteriophage replication which has an ssDNA genome and genetic recombination between genomes of λ bacteriophage (Red-recombination system).

When testing development of the M13 bacteriophage, we found that its efficiency is significantly more effective in the Δ*hfq* mutant relative to WT control (Fig. 5*a*). Since effects observed in mutants completely devoid of Hfq might be caused by the absence of various activities of this protein, we repeated these experiments using a

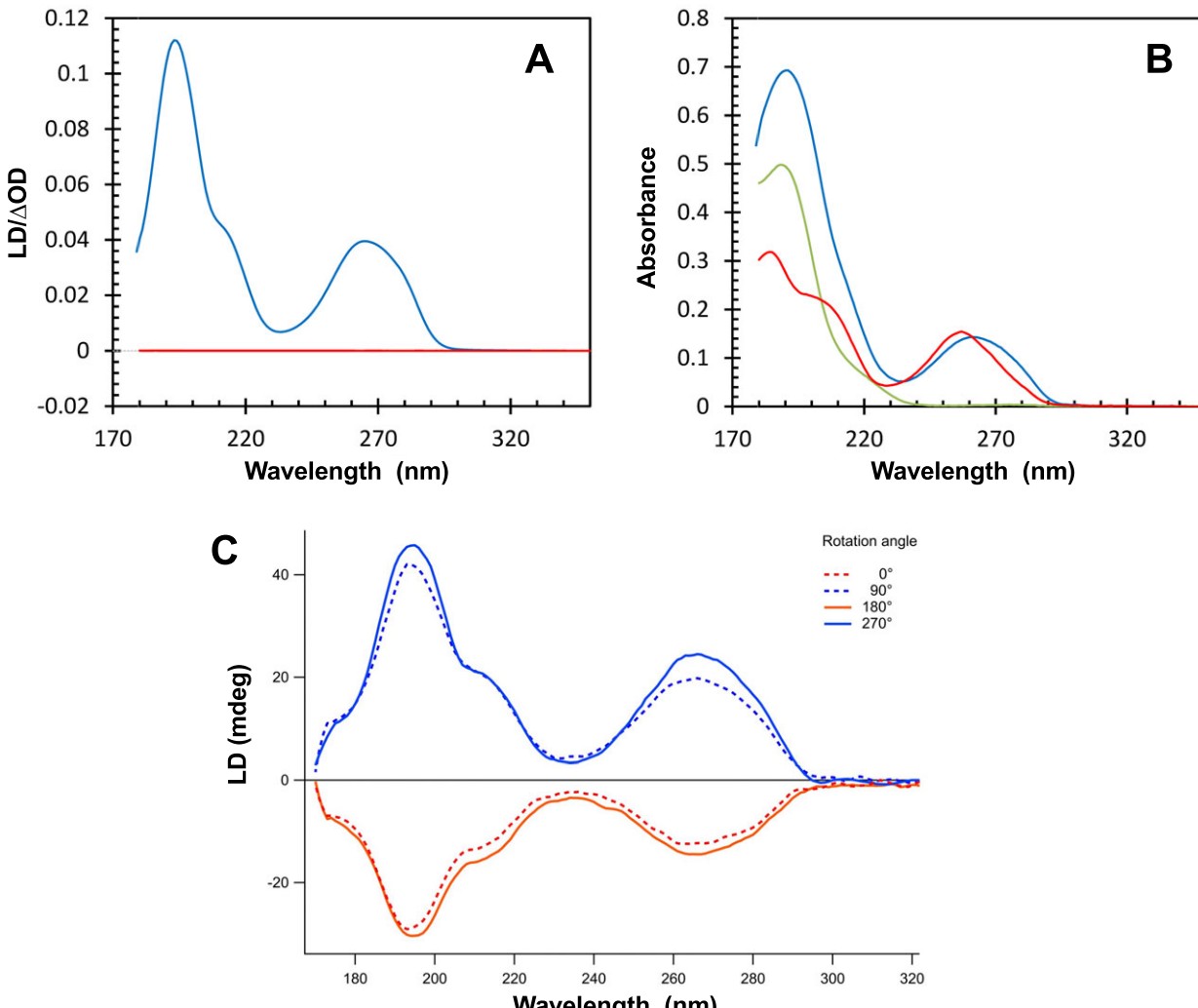

**Fig.4.** LD signal (*a*) and absorbance spectra (*b*) of the complex dA$_{59}$:CTR (blue), dA$_{59}$ (red) and Hfq-CTR (green). Spectra were measured with a sample path length of 0.5 mm and rotation speed of the Couette flow cell of 3000 rpm. (*c*) SRLD analysis of the same complex measured in a classic 0.024 mm path length cell, rotating the cell holder every 90°. The overall shape of the spectra is conserved with maxima and minima in the same positions compared to (*a*). Amplitude differences are most likely due to differences of the experiments, with a less perfect alignment of the sample in the classic cell.

mutant with deletion of the Hfq CTR while leaving intact the NTR. In such a mutant, replication of the M13 phage was significantly less efficient relative to the WT counterpart, as shown by slower phage development and lower burst size (Fig. 5*a*).

To test the efficiency of genetic recombination, we have employed λ bacteriophage mutants. Nonsense (amber, or shortly am) λ mutants in genes *P* or *S* can propagate only in suppressor *E. coli* hosts (*supE* or *supF*, respectively) but not in WT bacteria. When WT, Δ*hfq* or ΔCTR cells were infected simultaneously with both phage mutants, Red-recombination could occur. Culture growth was timely stopped and phage lysates were prepared after one single lytic development cycle was reached. Thus, phage progeny contained parental phages and recombinants, including λ*P*(am)*S*(am) and λ*P*$^+$*S*$^+$ variants. The latter phages could be easily distinguished from parental viruses because of their ability to form plaques on the WT host. By comparing phage titers on *WT* and *supE supF* strains, we found that the fraction of recombinant λ*P*$^+$*S*$^+$ phages after propagation in control (*hfq*$^+$) cells was equal to 0.17 ± 0.04%, which was significantly higher than fractions of

revertants (spontaneous *P*$^+$ and *S*$^+$ reverse mutants, which appear among mutant phages), achieving a frequency of 0.001% and < 0.0001% for λ*P*(am) and λ*S*(am), respectively. When testing efficiency of recombination in Δ*hfq* and ΔCTR hosts, evident influence of the absence of either whole Hfq or its CTR could be observed. Interestingly, the fraction of λ*P*$^+$*S*$^+$ recombinants was about 2-fold lower after propagation of phages in the Δ*hfq* host than in WT bacteria, while estimated efficiency of recombination was over two times higher in the ΔCTR host relative to the WT counterpart (Fig. 5*b*). Therefore, a lack of Hfq results in λ phage recombination inhibition, while in the presence of only Hfq-NTR, this process is more efficient in *E. coli*.

As the absence of the CTR could influence Hfq stability and its abundance (Arluison *et al.,* 2004), quantification of Hfq and its CTR-truncated form was performed; no significant differences were observed (Gaffke *et al.,* 2021). Therefore, we conclude that differences in efficiencies of phage M13 replication and phage λ genetic recombination between WT bacteria and *hfq* mutants arise from dysfunctions of Hfq in the mutant cells.

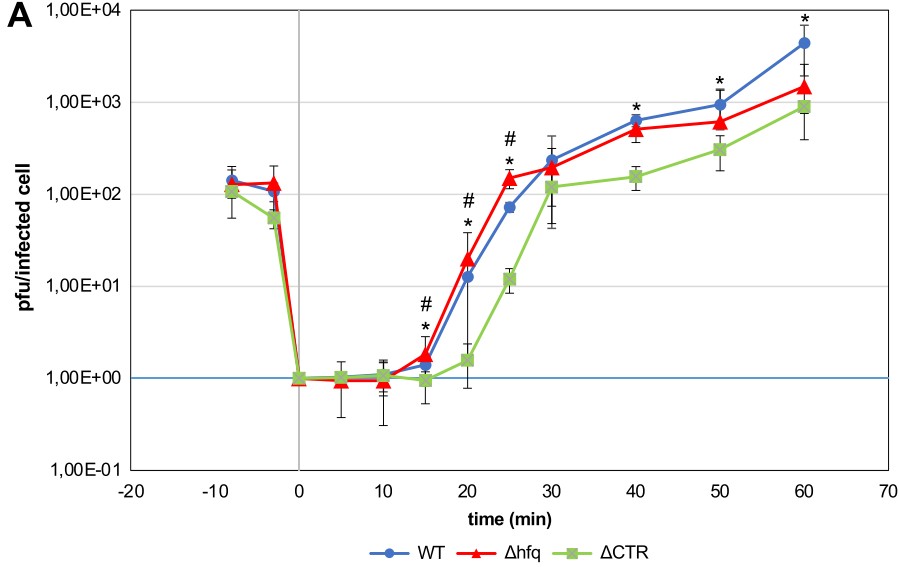

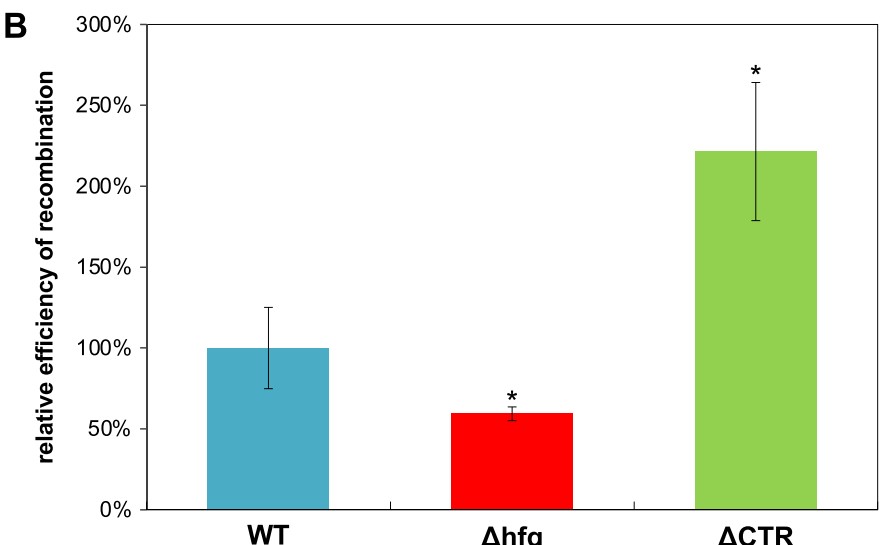

**Fig. 5.** (*a*) Development of bacteriophage M13 in *E. coli*. The presented results indicate mean values from three experiments with error bars indicating SD. Symbols (#) and (*) indicate statistically significant differences (*p* < 0.05 in the *t*-test) between results obtained for *hfq*$^+$ and Δ*hfq*, and *hfq*$^+$ and ΔCTR, respectively. (*b*) Efficiency of recombination between λ bacteriophage genomes in *E. coli*. The presented results show mean values from three experiments with error bars indicating SD. Value of 100% represents fraction of λ*P*$^+$*S*$^+$ recombinants appearing after infection of the *hfq*$^+$ host cells with λ*cI857S7*(am) and λ*b519imm21susP* phages which was equal to 0.17 ± 0.04%. Symbols (*) indicate statistically significant differences (*p* < 0.05 in the *t*-test) between results obtained for *hfq*$^+$ and either Δ*hfq* or ΔCTR hosts.

## Discussion

The results presented in this paper show that, in addition to its role in RNA metabolism, Hfq binds to ssDNA and may therefore play an important role in genetic processes involving ssDNA, including recombination and replication. *In vivo*, the Hfq protein has been discovered as a host factor for bacteriophage replication (Franze de Fernandez *et al.,* 1972). Further studies indicated then that this protein mediates various RNA transactions (Vogel and Luisi, 2011; Sobrero and Valverde, 2012).

Although early studies suggested that Hfq interacts with ssRNA, subsequent experiments indicated that this protein can interact with DNA (Cech *et al.,* 2016). In this report, we demonstrate that Hfq interacts also with ssDNA. Note that the role of Hfq in ssDNA binding and remodelling strengthens the general role of bacterial NAPs in similar processes, such as for HU and H-NS/StpA family (Shiraishi *et al.,* 2007; Kamashev *et al.,* 2008; Yang *et al.,* 2019).

We show here that the ssDNA-binding properties of Hfq are mainly due to its CTR. We observe that CTR binding to ssDNA allows its alignment by forming a parallel helix. This is not the case for ssDNA alone; thus, the CTR promotes the formation of this parallel helix probably by changing the p$K_A$ of adenines.

Interestingly, Hfq-CTR has also a propensity to juxtapose two DNA molecules and this is highly suggestive of a role in modulation of efficiency of recombination and/or transformation, as it was formerly shown for another "dsDNA bridging/ssDNA binding" protein, DprA (Mortier-Barriere *et al.,* 2007). To test if such interaction can play a physiological role we have performed efficiency assays for processes that require ssDNA. Our finding that two processes are affected, either positively (M13 replication) or negatively (λ recombination) in the Δ*hfq* mutant might suggest that this is the case. However, any secondary effects of Hfq-mediated changes in regulatory RNA functions could not be excluded in such

experimental systems. Therefore, we have also used the ΔCTR mutant in which only the NTR of Hfq is present. Surprisingly, we found that effects of the ΔCTR mutation on both M13 replication and λ recombination are more pronounced than in the case of Δ*hfq*, and opposite to those observed in cells completely lacking Hfq. These results corroborate the conclusion that ssDNA binding by Hfq has a physiological relevance. We have also demonstrated that *in vivo* Hfq-CTR stimulates replication of bacteriophage M13 genome, while inhibiting Red recombination. We propose that Hfq-mediated stimulation of M13-replication might relate to changes in ssDNA conformation facilitating alignment of ssDNA in parallel helices, and then the formation of the replication complex, similarly to the mechanism occurring in the Qβ bacteriophage replication. On the other hand, covering ssDNA regions by Hfq during genetic recombination may impair this process due to lower availability of recombining sequences during the exchange process of strands between two DNA molecules.

Recombination efficiency is also controlled by Hfq-CTR which inhibits recombination. Mutant Δ*hfq* cells lacking both the NTR and CTR regions recombine less efficiently in contrast to only CTR deficient cells. WT cells containing Hfq are not substantially more efficient in recombination than Δ*hfq*. Therefore, we hypothesise that effects of the absence of CTR in Δ*hfq* cells might be hidden due to the inability of the NTR to perform its functions. This includes interactions with sRNA and ultimately regulation of gene expression. This definitely makes Hfq an important player to consider in bacterial chromosome structure and gene expression control.

**Open Peer Review.** To view the open peer review materials for this article, please visit http://doi.org/10.1017/qrd.2022.15.

**Acknowledgements.** We are very grateful to C. Lavelle (MNHN, Paris), P. Dupaigne (IGR, Villejuif, France) and O. Pietrement (U. of Dijon, France) for their contribution to TEM measurements at an early stage of this work and for many fruitful discussions.

**Authors' contributions.** K.K. constructed plasmids and *E. coli* mutant strains and performed *in vivo* experiments (replication and recombination assessment). J.v.d.M. and I.Y. did optical microscopy of ssDNA complexes. F.W., V.A., N.C.J. and S.V.H. did SRCD, SRLD and OCD measurement and analysis. A.C. and E.L.C. did TEM measurements. F.G. did FTIR measurements. G.W. and V.A. concepted the study and supervised the work. All authors participated in the design, interpretation of the studies and analysis of the data and review of the manuscript.

**Supplementary Materials.** To view supplementary material for this article, please visit http://doi.org/10.1017/qrd.2022.15.

**Financial support.** This research was supported by Singapore Ministry of Education Academic Research Fund (Tier 1 R-144-000-451-114 and Tier 2 MOE-T2EP50121–0003 grants), National Science Center (Poland) (2016/21/N/NZ1/02850 to KK) and University of Gdansk (531-D020-D242–21 to GW). SRCD measurements on DISCO beamline at the SOLEIL Synchrotron were performed under proposal 20200007. SRLD measurements on ASTRID2 synchrotron radiation facility (Aarhus University, Denmark) were performed under proposal ISA-21-102. Campus France is gratefully acknowledged for their financial support of this work through PHC Polonium with Poland 27701VG. This study contributes to the IdEx Université de Paris ANR-18-IDEX-0001. This work was supported by a public grant overseen by the French National research Agency (ANR) as part of the 'Investissement d'Avenir' program, through the 'ADI 2019' project funded by the IDEX Paris-Saclay, ANR-11-IDEX-0003-02.

**Conflict of interest.** The authors declare no potential conflicts of interest with respect to the research, authorship and/or publication of this article.

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
