## [Reviewer Report]

*Comments to Author*: 1. The general reader would benefit from the inclusion of drawings (a new Figure 1?) summarizing the structure of the Hfq hexamer while in contact with nucleic acids. This would allow the contributions of the NTR and the CTR to nucleic acid interaction to be appreciated more easily.

2. Lines 80-81. In order to provide a more comprehensive, comparative survey of Hfq and other RNA binding proteins from Escherichia coli, please cite Rajkowitsch L and Schroeder R (42) Dissecting RNA chaperone activity. RNA 13: 2053-2060.

3. Lines 107-109. "If Hfq does not affect DNA supercoiling and transcription directly it

4. possibly regulates them indirectly, for instance by a transcriptional regulator expression (Malabirade et al., 2018)." Please clarify the meaning of "by a transcriptional regulator expression". Do you mean: by affecting the expression of a transcription regulator?

5. Line 122. We determined that the pH~5 used was the most appropriate to form the complex with DNA. This is not the physiological pH of the bacterial cytosol (which is closer to neutral pH). Given that the CTR fragment of Hfq is being discussed, please comment on the pH discrepancy in terms of physiological relevance of the acidic pH value. The point is relevant because pathogenic strains of Gram-negative bacteria can acidify the cytosol when adapting to the vacuole of the host macrophage and many virulence genes belong to the Hfq regulon in these bacterial species.

6. Line 164. In what way is E. coli strain MG1655 an hfq+ 'variant'? Do you mean that it is a positive control for the hfq and CTR deletion mutants?

7. Lines 181-186. The selection of the 59-mer A-tract was made on the grounds of experimental convenience. It would be helpful to link this selection to the in vivo situation by citing examples of known Hfq binding regions in bacteria with this sequence. Is there even one biological example?

8. Lines 223-229. The experiments described here seem to involve Phi-X, which is described variously as a 'virion' and a 'plasmid' in the legend to Figure 1C (lines 552-557). This element was not introduced in the Materials and Methods section devoted to bacteriophage (lines 163-176). Please describe Phi-X and carefully define the terms 'virion' and 'plasmid' as used by the authors in this manuscript.

9. Lines 220-229. Intra- and intermolecular bridging is introduced. This topic should be summarized in the Introduction, and the paper by Rajkowitsch & Schroeder (42) RNA 13: 2053-2060, cited.

10. The finding that Hfq binds, coats and spreads ssDNA is perhaps not surprising (lines 180-229). However, it is interesting that the CTR and NTR do the same. The CD experiments (lines 231-290) reveal that Hfq promotes alignment of A-tracts and that these have a B-form structure (line 250).

11. The conclusions drawn in lines 283-290 about the SSB-like nature of Hfq are based on a limited number of experiments with a 59-mer A-tract studied at pH 5 in vitro. Given that Hfq has a well-established biological role as an RNA chaperone, such conclusions about its wider role(s) in the bacterium must await more experimentation. At present, it would be safer to characterize these as speculation rather than firm conclusions.

12. Lines 292-325. Biological experiments with bacteriophage revealed effects of Hfq and its CTR on molecular processes associated with lambda and M13 phage. However, no mechanistic insights have emerged so far, so we are left with just generalized observations about correlations between Hfq, its CTR and the life cycles of two phages.

13. The role of Hfq in ssDNA binding and modeling is reminiscent of the role of bacterial nucleoid-associated protein HU in similar processes (Kamashev et al. [26]. Nucleic Acids Research 36: 1026-1036). HU has roles in both RNA and DNA metabolism, in the latter case, interacting with both ds- and ssDNA. This provides an interesting point of comparison between two heavily investigated proteins involved in genome metabolism. Some commentary on this topic would add to the scholarly value of the present paper.

14. The authors have used sophisticated in vitro methods to study a difficult system. They have been very frank in describing the limits of their findings. Despite the general lack of specific, mechanistic insights at the present time, the work does provide evidence that the ssDNA binding activity of Hfq has physiological relevance at some level. Therefore, it provides a platform for deeper analysis of these molecular processes in the future.

---

## [Reviewer Report]

*Comments to Author*: Dear Dr. Arluison,we have obtained one very detailed review of your manuscript but have had difficulty obtaining another.This reviewer is quite positive but suggests a number of changes/additions.Please respond to each comment and indicate changes, if any, that you have made.

Reviewer #1: 1. The general reader would benefit from the inclusion of drawings (a new Figure 1?) summarizing the structure of the Hfq hexamer while in contact with nucleic acids. This would allow the contributions of the NTR and the CTR to nucleic acid interaction to be appreciated more easily.

2. Lines 80-81. In order to provide a more comprehensive, comparative survey of Hfq and other RNA binding proteins from Escherichia coli, please cite Rajkowitsch L and Schroeder R (42) Dissecting RNA chaperone activity. RNA 13: 2053-2060.

3. Lines 107-109. "If Hfq does not affect DNA supercoiling and transcription directly it

4. possibly regulates them indirectly, for instance by a transcriptional regulator expression (Malabirade et al., 2018)." Please clarify the meaning of "by a transcriptional regulator expression". Do you mean: by affecting the expression of a transcription regulator?

5. Line 122. We determined that the pH~5 used was the most appropriate to form the complex with DNA. This is not the physiological pH of the bacterial cytosol (which is closer to neutral pH). Given that the CTR fragment of Hfq is being discussed, please comment on the pH discrepancy in terms of physiological relevance of the acidic pH value. The point is relevant because pathogenic strains of Gram-negative bacteria can acidify the cytosol when adapting to the vacuole of the host macrophage and many virulence genes belong to the Hfq regulon in these bacterial species.

6. Line 164. In what way is E. coli strain MG1655 an hfq+ 'variant'? Do you mean that it is a positive control for the hfq and CTR deletion mutants?

7. Lines 181-186. The selection of the 59-mer A-tract was made on the grounds of experimental convenience. It would be helpful to link this selection to the in vivo situation by citing examples of known Hfq binding regions in bacteria with this sequence. Is there even one biological example?

8. Lines 223-229. The experiments described here seem to involve Phi-X, which is described variously as a 'virion' and a 'plasmid' in the legend to Figure 1C (lines 552-557). This element was not introduced in the Materials and Methods section devoted to bacteriophage (lines 163-176). Please describe Phi-X and carefully define the terms 'virion' and 'plasmid' as used by the authors in this manuscript.

9. Lines 220-229. Intra- and intermolecular bridging is introduced. This topic should be summarized in the Introduction, and the paper by Rajkowitsch & Schroeder (42) RNA 13: 2053-2060, cited.

10. The finding that Hfq binds, coats and spreads ssDNA is perhaps not surprising (lines 180-229). However, it is interesting that the CTR and NTR do the same. The CD experiments (lines 231-290) reveal that Hfq promotes alignment of A-tracts and that these have a B-form structure (line 250).

11. The conclusions drawn in lines 283-290 about the SSB-like nature of Hfq are based on a limited number of experiments with a 59-mer A-tract studied at pH 5 in vitro. Given that Hfq has a well-established biological role as an RNA chaperone, such conclusions about its wider role(s) in the bacterium must await more experimentation. At present, it would be safer to characterize these as speculation rather than firm conclusions.

12. Lines 292-325. Biological experiments with bacteriophage revealed effects of Hfq and its CTR on molecular processes associated with lambda and M13 phage. However, no mechanistic insights have emerged so far, so we are left with just generalized observations about correlations between Hfq, its CTR and the life cycles of two phages.

13. The role of Hfq in ssDNA binding and modeling is reminiscent of the role of bacterial nucleoid-associated protein HU in similar processes (Kamashev et al. [26]. Nucleic Acids Research 36: 1026-1036). HU has roles in both RNA and DNA metabolism, in the latter case, interacting with both ds- and ssDNA. This provides an interesting point of comparison between two heavily investigated proteins involved in genome metabolism. Some commentary on this topic would add to the scholarly value of the present paper.

14. The authors have used sophisticated in vitro methods to study a difficult system. They have been very frank in describing the limits of their findings. Despite the general lack of specific, mechanistic insights at the present time, the work does provide evidence that the ssDNA binding activity of Hfq has physiological relevance at some level. Therefore, it provides a platform for deeper analysis of these molecular processes in the future.

---

## [Reviewer Report]

*Comments to Author*: Dear Dr. Arluison,Thanks very much for response to the reviewer and the changes made to the manuscript.With these changes, your manuscript is accepted for publication.